# Heterozygous deletion of Gpr55 does not affect a hyperthermia-induced seizure, spontaneous seizures or survival in the $Scn1a^{+/-}$ mouse model of Dravet syndrome

Lyndsey L. Anderson[1,2,3], Dilara A. Bahceci[1,2,3], Nicole A. Hawkins[4], Declan Everett-Morgan[3], Samuel D. Banister[1,3,5], Jennifer A. Kearney[4], Jonathon C. Arnold [1,2,3]*

1 Brain and Mind Centre, The University of Sydney, Sydney, NSW, Australia, 2 Faculty of Medicine and Health, Discipline of Pharmacology, Sydney Pharmacy School, The University of Sydney, Sydney, NSW, Australia, 3 Lambert Initiative for Cannabinoid Therapeutics, The University of Sydney, Sydney, NSW, Australia, 4 Department of Pharmacology, Feinberg School of Medicine, Northwestern University, Evanston, IL, United States of America, 5 Faculty of Science, School of Chemistry, The University of Sydney, Sydney, NSW, Australia

* jonathon.arnold@sydney.edu.au

## Abstract

A purified preparation of cannabidiol (CBD), a cannabis constituent, has been approved for the treatment of intractable childhood epilepsies such as Dravet syndrome. Extensive pharmacological characterization of CBD shows activity at numerous molecular targets but its anticonvulsant mechanism(s) of action is yet to be delineated. Many suggest that the anticonvulsant action of CBD is the result of G protein-coupled receptor 55 (GPR55) inhibition. Here we assessed whether Gpr55 contributes to the strain-dependent seizure phenotypes of the $Scn1a^{+/-}$ mouse model of Dravet syndrome. The $Scn1a^{+/-}$ mice on a 129S6/SvEvTac (129) genetic background have no overt phenotype, while those on a [129 x C57BL/6J] F1 background exhibit a severe phenotype that includes hyperthermia-induced seizures, spontaneous seizures and reduced survival. We observed greater Gpr55 transcript expression in the cortex and hippocampus of mice on the seizure-susceptible F1 background compared to those on the seizure-resistant 129 genetic background, suggesting that Gpr55 might be a genetic modifier of $Scn1a^{+/-}$ mice. We examined the effect of heterozygous genetic deletion of Gpr55 and pharmacological inhibition of GPR55 on the seizure phenotypes of F1. $Scn1a^{+/-}$ mice. Heterozygous Gpr55 deletion and inhibition of GPR55 with CID2921524 did not affect the temperature threshold of a thermally-induced seizure in F1.$Scn1a^{+/-}$ mice. Neither was there an effect of heterozygous Gpr55 deletion observed on spontaneous seizure frequency or survival of F1.$Scn1a^{+/-}$ mice. Our results suggest that GPR55 antagonism may not be a suitable anticonvulsant target for Dravet syndrome drug development programs, although future research is needed to provide more definitive conclusions.

**Data Availability Statement:** All relevant data are within the paper and its Supporting Information files.

**Funding:** This research was supported by the Lambert Initiative for Cannabinoid Therapeutics, a philanthropically-funded centre for medicinal cannabis research at the University of Sydney, the Australian National Health and Medical Research Council (GNT1161571) and the U.S. National Institutes of Health (R01 NS084959). The Monash Genome Modification Platform, Monash University is a node of Phenomics Australia, which is supported by the Australian Government Department of Education through the National Collaborative Research Infrastructure Strategy, the Super Science Initiative and Collaborative Research Infrastructure Scheme.

**Competing interests:** The authors declare that the research was conducted in the absence of any commercial or financial relationships that could be construed as a potential conflict of interest. We confirm that this does not alter our adherence to all PLOS ONE policies on sharing data and materials.

## Introduction

Cannabidiol (CBD) and cannabis-based products are frequently being used as anticonvulsants in epilepsy patients [1]. Drug regulatory agencies in the United States, Europe and Australia have approved a purified preparation of CBD, Epidiolex®, for the treatment of the intractable childhood epilepsies Dravet syndrome and Lennox-Gastaut syndrome. Despite vast research detailing the anticonvulsant effects of CBD and other cannabinoids, the anti-seizure mechanism(s) of action of this compound class is poorly understood (Gray and Whalley, 2020) [2].

One favored mechanism to account for the anticonvulsant effects of CBD is inhibition of G protein-coupled receptor 55 (GPR55) [2,3]. GPR55 was once suggested to be the third cannabinoid receptor, as endocannabinoids activated the receptor at low nanomolar concentrations [4]. Data since suggests that the sphingolipid lysophosphatidlyinositol (LPI) is more likely to be the endogenous ligand for GPR55 [5,6]. GPR55 is coupled to $G\alpha_{12/13}$ and $G\alpha_q$ and influences intracellular signalling mechanisms, including calcium release from intracellular stores, stimulation of extracellular signal regulated kinase (ERK1/2), and activation of transcription factors, such as nuclear factor of activated T cells (NFAT) and cAMP response element binding protein (CREB) [4,5,7–9]. Within the central nervous system, GPR55 is expressed on hippocampal pyramidal cells where its activation promotes neurotransmitter release and neuronal excitation [10,11].

Cannabinoids with anticonvulsant properties, such as CBD, $\Delta^9$-THC, cannabidiolic acid (CBDA), cannabigerolic acid (CBGA), cannabidivarin (CBDV) and $\Delta^9$-tetrahydrocannabivarin ($\Delta^9$-THCV), all inhibit GPR55 [4,12–17]. The anticonvulsant action of CBD in a mouse model of Dravet syndrome was attributed to antagonism of GPR55, where CBD and the selective GPR55 antagonist CID16020064 [18] similarly restored loss of inhibitory neurotransmission in the dentate gyrus and CID16020064 blocked the effects of CBD [3]. Collectively, these data suggest that inhibition of GPR55 might provide a new anticonvulsant drug target for intractable childhood epilepsies.

Dravet syndrome is a treatment-resistant epilepsy that typically presents in infants as febrile seizures that progress to multiple afebrile seizure types [19]. Loss-of-function mutations in *SCN1A*, the gene that encodes $Na_v1.1$, are present in the majority of Dravet syndrome patients [19,20]. Heterozygous deletion of *Scn1a* (*Scn1a$^{+/-}$*) in mice reproduces the core clinical features of Dravet syndrome, including hyperthermia-induced seizures, spontaneous seizures and poor survival [21,22]. Penetrance of these phenotypes, however, is strain-dependent [23–25]. *Scn1a$^{+/-}$* mice on a congenic 129S6/SvEvTac strain (129.*Scn1a$^{+/-}$*) are seizure-resistant, with no overt epilepsy phenotype. F1.*Scn1a$^{+/-}$* mice, generated by crossing 129.*Scn1a$^{+/-}$* mice with wild-type C57BL/6J mice, are seizure-susceptible and exhibit a severe Dravet syndrome phenotype. This divergence in epilepsy phenotype depending on the mouse background strain suggests that other factors, such as genetic modifiers, affect disease severity. Genetic modifiers are genes distinct from the primary mutation that influence the disease phenotype, which could serve as novel drug targets [26]. RNA-seq analysis of an *Scn1a$^{+/-}$* mouse model of Dravet syndrome identified *Gpr55* as a candidate genetic modifier [27].

The present study aimed to determine whether Gpr55 affects the epilepsy phenotype of F1.*Scn1a$^{+/-}$* mice to infer its potential as a new drug target for the treatment of Dravet syndrome. First, we compared cortical and hippocampal *Gpr55* transcript expression between seizure-susceptible (F1) and seizure-resistant (129) genetic background strains. We then determined whether heterozygous deletion of *Gpr55* is anticonvulsant in the F1.*Scn1a$^{+/-}$* mouse model of Dravet syndrome. We also identified the Gpr55 inhibitor CID2921524 as brain-penetrant and examined its anticonvulsant potential in F1.*Scn1a$^{+/-}$* mice.

## Materials and methods

### Animals

All animal care and procedures were approved by the University of Sydney Animal Ethics Committee in accordance with the Australian Code of Practice for the Care and Use of Animals for Scientific Purposes or the Northwestern University Animal Care and Use Committees in accordance with the National Institutes of Health Guide for the Care and Use of Laboratory Animals. Mice were group-housed in specific pathogen-free mouse facilities under standard conditions with *ad libitum* access to food and water. Gene expression studies were conducted at Northwestern University (Chicago, USA) where the mouse facility operated under a 14 h light/10 h dark cycle. All other studies were conducted at the University of Sydney (Sydney, AUS) where the mouse facility operated under a 12 h light/12 h dark cycle.

### $Scn1a^{+/-}$ mice

Mice heterozygous for *Scn1a* ($Scn1a^{+/-}$) were generated by targeted deletion of exon 1 and maintained as a congenic line on the 129S6/SvEvTac background ($129.Scn1a^{+/-}$) as described [24]. For studies conducted at the University of Sydney, $129.Scn1a^{+/-}$ mice were purchased from The Jackson Laboratory (stock 37107-JAX; Bar Harbor, USA). C57BL/6J (wildtype or $Gpr55^{+/-}$) mice were bred with heterozygous $129.Scn1a^{+/-}$ mice to generate [129 x B6]F1 mice. The *Scn1a* genotype was determined as previously described [24].

### $Gpr55^{+/-}$ mice

Mice heterozygous for *Gpr55* ($Gpr55^{+/-}$) on the C57BL/6J inbred strain were generated using CRISPR genome editing by Monash Genome Modification Platform (Monash University; Melbourne, AUS). The UCSC Genome Browser (https://genome.ucsc.edu) was used to identify guide RNA target sites flanking the exon ENSMUST00000086975.5 of *Gpr55*. CRISPR-Cas9 guide RNA was ordered from Integrated DNA Technologies (Coralville, USA) with 5' to 3' sequences of CATTAAGCATGGGTCGCTAA (sgRNA1) and GCTGGTCAACCAGGATACCA (sgRNA2). C57BL/6J zygotes were electroporated with Alt-R® S.p. Cas9 Nuclease V3 (3.8 μM) and sgRNAs (4.2 μM). Electroporated zygotes were transferred to the uterus of pseudopregnant females. The *Gpr55* genotype was determined by PCR on DNA extracted from tail biopsies. DNA was isolated from tail biopsies using the Gentra Puregene Mouse Tail Kit according to the manufacturer's instructions (Qiagen; Valencia, USA). A multiplex PCR using a forward primer located upstream of the deletion and reverse primers located within and downstream of the deletion was used. Genotyping primer 5' to 3' sequences were as follows: AGGCTCGTGCACAGAAGAG (F), AAGCCTCGGATGGCCAGTAG (WT-R) and CCCTAGCCCT‒GAATCACCACA (KO-R). PCR conditions were 94˚C for 2 min, then 36 cycles of 94˚C for 20 s, 60˚C for 20 s and 72˚C for 2 min before a final step of 72˚C for 5 min. These primers amplify a 673 bp product from the wildtype allele and a 785 bp product from the targeted allele. $Gpr55^{+/-}$ mice were maintained as a congenic line on the C57BL/6J background.

### Pharmacokinetic analysis

A pharmacokinetic study was conducted for CID2921524 (MolPort; Riga, LVA) to determine the target experimental time point for the hyperthermia-induced seizure protocol. Male and female wildtype mice (P21-28) received a single intraperitoneal (i.p.) injection of 10 mg/kg CID2921524 in ethanol-Tween 80-saline (1:1:18) with an injection volume of 10 mL/kg. At selected time points, mice were anaesthetized with isoflurane and whole blood was collected

via cardiac puncture. Plasma was isolated by centrifugation, whole brain was also collected and samples were stored at -80˚C until assayed.

Plasma and brain samples were prepared for analysis of CID2921524 concentrations as previously described [28]. Samples were assayed by LC-MS/MS using a Shimadzu Nexera ultra-HPLD coupled to a Shimadzu 8030 triple quadrupole mass spectrometer (Shimadzu Corp.; Kyoto, JPN) operated in positive electrospray ionization mode with multiple reaction monitoring with 376.3 > 121.1 and 376.2 > 77.0 mass transition pairs. Quantification was achieved by comparing experimental samples to standards prepared with known amounts of drug.

## Hyperthermia-induced seizures

Hyperthermia-induced seizure experiments were conducted on male and female F1.$Scn1a^{+/-}$ and F1.$Scn1a^{+/-}$;$Gpr55^{+/-}$ mice at P14-16 as previously described [21]. Briefly, a RET-3 rectal temperature probe was inserted and mice acclimated for 5 min before mouse core body temperature was elevated 0.5˚C every 2 min until the onset of first clonic convulsion with loss of posture or until 42.5˚C was reached. For hyperthermia-induced seizure experiments with CID2921524, mice received a single i.p. injection of vehicle or CID2921524 at an injection volume of 10 mL/kg following the 5 min acclimation period.

## Spontaneous seizures and survival

Male and female F1.$Scn1a^{+/-}$ and F1.$Scn1a^{+/-}$;$Gpr55^{+/-}$ mice were exposed to a single hyperthermia-induced seizure event at P18 as described previously [21]. Mice were housed in groups of three, continuous video recordings were captured for 60 h and spontaneous generalized tonic-clonic seizures (GTCS) from 12:00 P19 to 24:00 P21 were quantified and scored as previously described [21]. Mice were monitored daily to P30 to assess survival. Dead mice were promptly removed from the homecages. Human endpoints cannot be used in the measurement of survival, as the deaths that occur are spontaneous and cannot be predicted. There is thus no alternative means to measure survival. Spontaneous mortality is an important measure and models the human phenomenon of sudden unexpected death in epilepsy (SUDEP) that occurs in patients with Dravet syndrome. The anticipated mortality was approved by the University of Sydney's Animal Ethics Committee. Out of 70 mice tested, 42 mice died in the survival monitoring period.

## Quantitative reverse transcription droplet digital PCR (RT-ddPCR)

Transcript expression of target genes was determined using RT-ddPCR as previously described [27]. Briefly, cortex and hippocampi were dissected from mice at postnatal day 24 (P24) and tissue from 3–4 mice (at least one from each sex) were combined into pooled samples to isolate total RNA. Cortical and hippocampal samples were dissected and pooled from different cohorts of F1.$Scn1a^{+/-}$ mice. First strand cDNA was synthesized (Superscript IV; Life Technologies; Carlsbad, CA, USA) from 2 μg total RNA and ddPCR was performed using ddPCR Supermix for Probes (No dUTP; Bio-Rad; Hercules, CA, USA) and TaqMan Assays as previously described [23] on n = 7–11 pooled samples per group. Taqman gene expression assays (Life Technologies) were mouse $Gpr55$ (FAM-MGB-Mm02621622_s1) and $Tbp$ (VIC-MGB-Mm00446971_m1). Relative transcript levels were expressed as a concentration ratio to Tbp.

## Immunoblotting

Brain protein was isolated from adult wildtype and $Gpr55^{-/-}$ mice generously provided by Dr Kenneth Mackie (Indiana University; Bloomington, USA). Mice were sacrificed at P24 by

cervical dislocation and immediately decapitated. Whole brains were extracted and immediately frozen with liquid nitrogen and stored at -80˚C. For dissections, whole brain samples were thawed on ice before the hippocampi from each hemisphere was extracted and frozen with liquid nitrogen and stored at -80˚C. Western blot analysis was performed on membrane proteins that were isolated by differential centrifugation. Membrane proteins (100 μg) were separated on a 10% SDS-PAGE gel and transferred to a PVDF membrane. Proteins were detected with primary antibodies directed against GPR55 [ThermoFisher Scientific (720285; Waltham, USA), Abcam (ab203663; Cambridge, GBR) and Cayman Chemical (10224; Ann Arbor, USA)], β-tubulin (mouse; anti-β-tubulin monoclonal; 1:500; T5201; Sigma-Aldrich; St. Louis, USA) or β-actin (mouse; anti-β-actin monoclonal; 1:500; A5316; Sigma-Aldrich). Immunoreactive bands were detected with an Odyssey imager (LI-COR Biosciences; Lincoln, USA) using fluorescent secondary antibodies directed at the primary antibodies (goat:anti-rabbit 800 or goat:anti-mouse 680; 1:20,000; ThermoFischer Scientific). The Cayman primary anti-GPP55 antibody (1:200) was incubated with 7.5 μg GPR55 blocking peptide (10225; Cayman Chemical) for 1 h prior to immunoblotting.

## Data analyses

Normality of each dataset was assessed using the Shapiro-Wilk test. Concentrations of CID2921524 in plasma and brain at each time point were averaged and pharmacokinetic parameters were calculated by noncompartmental analysis as previously described [28]. Seizure threshold temperatures were compared using Mantel-Cox logrank test. Spontaneous seizure data were analysed using two-way ANOVA (seizure frequency and seizure severity), Fisher's exact test (proportion of mice seizure-free) or Mantel-Cox logrank test (survival). RT-ddPCR data were analysed using two-way ANOVA. All results $p < 0.05$ were considered statistically significant.

## Results

### Greater *Gpr55* mRNA expression in seizure-susceptible mice

A strain-dependent phenotype is observed for $Scn1a^{+/-}$ mice with 129.$Scn1a^{+/-}$ mice having no overt phenotype and F1.$Scn1a^{+/-}$ mice exhibiting a severe epilepsy phenotype. We compared the mRNA expression of *Gpr55* across mouse background strains to assess whether Gpr55 could be contributing to the strain-dependent effects. Transcript expression of *Gpr55* was examined in the cortex and hippocampus (Fig 1). Strain-dependent effects were observed in both the cortex and hippocampus, with wildtype and $Scn1a^{+/-}$ mice on the seizure-susceptible [129 x B6]F1 background strain having significantly greater *Gpr55* mRNA expression than mice on the seizure-resistant 129 background strain (cortex: $F_{1,26} = 82.20$, $p < 0.0001$; hippocampus: $F_{1,31} = 63.10$, $p < 0.0001$; two-way ANOVA). In the cortex, a strain by genotype interaction effect ($F_{1,26} = 7.212$, $p = 0.0124$; two-way ANOVA) was observed where $Scn1a^{+/-}$ mice had significantly greater *Gpr55* expression than wildtype mice on the [129 x B6]F1 background strain ($p = 0.0178$). These results suggest that *Gpr55* could be a genetic modifier of $Scn1a^{+/-}$ mice. Unfortunately, we were unable to determine whether the differences in mRNA expression translated to differences in Gpr55 protein expression. Three commercially available GPR55 antibodies (ThermoFisher, Abcam and Cayman Chemical) were not selective for Gpr55 in Western blots of mouse whole brain lysates (S1 Fig). Despite being unable to confirm whether the increased transcript expression translated to an increased Gpr55 protein expression, we sought to determine whether the increased *Gpr55* expression contributes to the seizure phenotype of F1.$Scn1a^{+/-}$ mice.

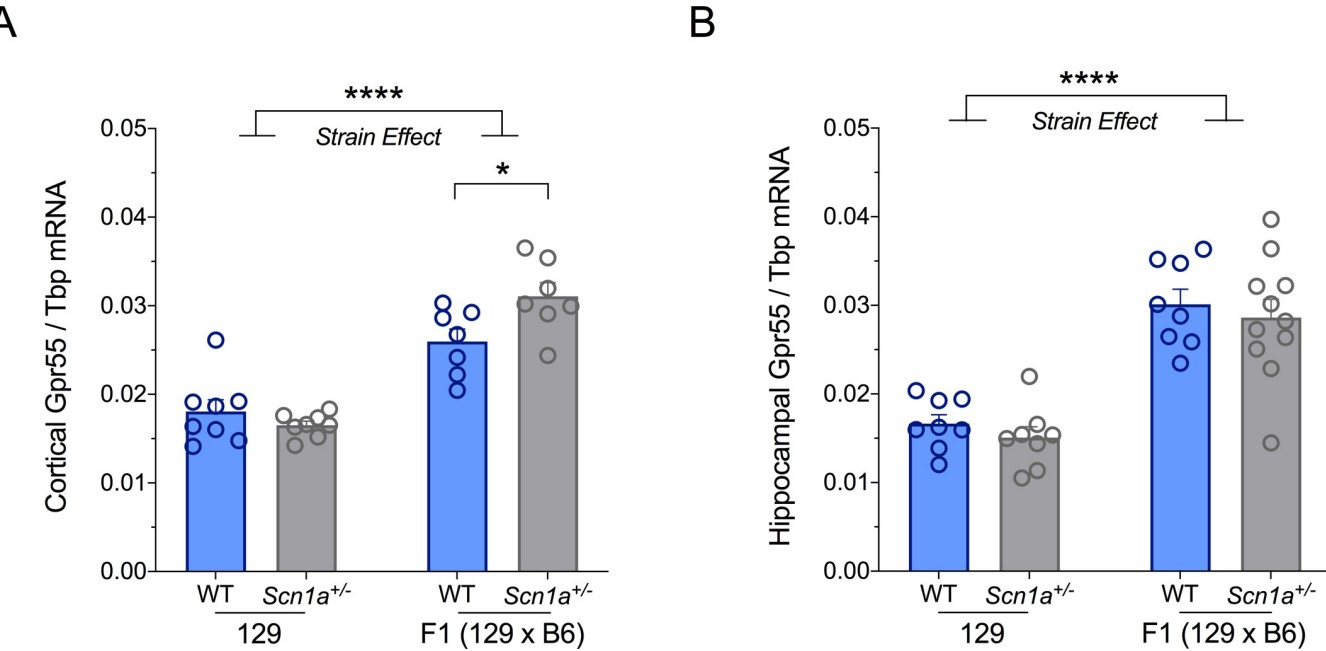

**Fig 1. Strain-specific *Gpr55* expression.** Relative *Gpr55* transcript levels in the (**A**) cortex and (**B**) hippocampus of wildtype (WT, blue bars) and *Scn1a*[+/-] (gray bars) mice on 129 and F1 (129 x B6) background strains. *Gpr55* expression was measured in primary RNA pools using RT-ddPCR and are expressed as a ratio of *Tbp*. Each primary pool includes tissue from 3–4 mice, with at least one from each sex. Data represents mean ± SEM, with n = 7–11 pools per group. Cortical and hippocampal samples were dissected and pooled from different cohorts of F1.*Scn1a*[+/-] mice. Cortical and hippocampal *Gpr55* expression was significantly greater in F1 compared to 129 mice (****$p < 0.0001$, two-way ANOVA). *Scn1a*[+/-] mice expressed significantly greater *Gpr55* than wildtype mice on the F1 (129 x B6) background strain (*$p < 0.05$ two-way ANOVA followed by Sidak's test).

## Generation of *Gpr55*[+/-] mice

CRISPR/Cas9 genome editing was used to generate *Gpr55* knockout mice on a congenic C57BL/6J background (Fig 2A). We generated a founder with a 1122 bp genomic deletion that covers the single exon of *Gpr55*. Heterozygous breeding pairs were used to confirm deletion of *Gpr55* with genotypes of wildtype, *Gpr55*[+/-] and *Gpr55*[-/-] mice (Fig 2B). Because a specific Gpr55 antibody is not available, we were unable to show that *Gpr55* deletion results in no

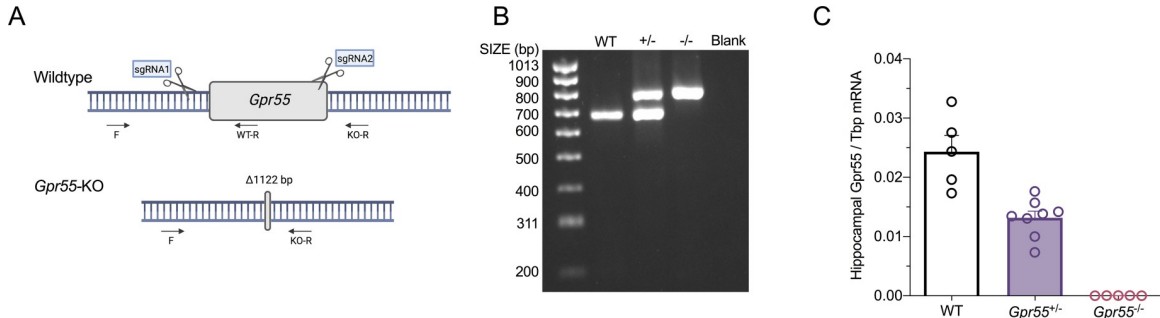

**Fig 2. Generation and molecular characterization of *Gpr55*[+/-] mice.** (**A**) Schematic of the CRISPR/Cas9 strategy to delete *Gpr55*. The binding sites of the CRISPR/Cas9 guide RNAs are noted as sgRNA1 and sgRNA2. The arrows show binding sites of the genotyping primers. Schematic created using BioRender.com. (**B**) DNA genotyping gel showing wildtype (WT), *Gpr55*[+/-] (+/-) and *Gpr55*[-/-] progeny (-/-) with a no DNA control (Blank). (**C**) Relative *Gpr55* transcript levels in the hippocampus of wildtype, *Gpr55*[+/-] and *Gpr55*[-/-] mice, revealing a complete loss of *Gpr55* expression in homozygous knockouts. *Gpr55* expression was measured in primary RNA pools using RT-ddPCR and expressed as a ratio of *Tbp*. Data represents mean ± SEM, with n = 5–8 per group (WT: 3 males, 2 females; *Gpr55*[+/-]: 4 males, 4 females; *Gpr55*[-/-]: 2 males, 3 females).

Gpr55 protein expression. However, RT-ddPCR was used to evaluate *Gpr55* transcript expression across all three genotypes (Fig 2C). An allele dose-response was observed for *Gpr55* expression ($F_{2,15} = 49.37$, $p < 0.0001$; one-way ANOVA). The absence of *Gpr55* mRNA in *Gpr55*$^{-/-}$ mice confirms that *Gpr55* was indeed deleted from the genome.

## *Gpr55* deletion has little effect on seizure phenotypes of *Scn1a*$^{+/-}$ mice

In order to assess whether the increased *Gpr55* expression of F1.*Scn1a*$^{+/-}$ mice contributes to the Dravet syndrome phenotype, we generated F1.*Scn1a*$^{+/-}$ mice with heterozygous deletion of *Gpr55* (F1.*Scn1a*$^{+/-}$;*Gpr55*$^{+/-}$). We first examined the effect of *Gpr55* deletion on a hyperthermia-induced seizure, which models febrile seizures that occur in children with Dravet syndrome (Fig 3A). Heterozygous deletion of *Gpr55* had no effect on a thermally induced seizure at P14-16 as the temperature threshold for GTCS of F1.*Scn1a*$^{+/-}$;*Gpr55*$^{+/-}$ mice (40.4 ± 0.1°C) was not different from that of F1.*Scn1a*$^{+/-}$ mice (40.2 ± 0.1°C, $p = 0.5190$, Mantel-Cox logrank test). Sex did not affect hyperthermia-induced seizure temperature thresholds, so males and females were combined.

Next, we assessed the effect of *Gpr55* deletion on spontaneous seizure frequency and severity of F1.*Scn1a*$^{+/-}$ mice. Since Dravet syndrome patients typically present with febrile seizures that then progress to spontaneous afebrile seizures, we model this progression by priming F1.*Scn1a*$^{+/-}$ mice with a hyperthermia-induced seizure at P18 and then measuring subsequent spontaneous seizure frequency (Fig 3B). Again, neither sex nor *Gpr55* genotype had any effect on the GTCS temperature threshold of the priming seizure event (F1.*Scn1a*$^{+/-}$: 40.3 ± 0.1°C vs. F1.*Scn1a*$^{+/-}$;*Gpr55*$^{+/-}$: 40.2 ± 0.1°C, $p = 0.4381$, Mantel-Cox logrank test). Interestingly, a sex-dependent effect on spontaneous seizure frequency was observed with female mice exhibiting a significantly higher seizure frequency than male mice ($F_{1,65} = 8.760$, $p = 0.0043$; two-way ANOVA). Spontaneous GTCS frequency was greater in female F1.*Scn1a*$^{+/-}$;*Gpr55*$^{+/-}$ mice than male F1.*Scn1a*$^{+/-}$;*Gpr55*$^{+/-}$ mice ($p = 0.0081$). Given there were less female mice than male mice tested and some of the female mice appeared as potential outliers, the robustness of this finding could be open to question. However, the severity of spontaneous GTCS as measured by the percentage of seizures that progress to the most severe stage of full tonic hindlimb extension was not affected by sex, so male and female data was combined. Heterozygous deletion of *Gpr55* did not impact spontaneous GTCS severity of F1.*Scn1a*$^{+/-}$ mice (Fig 3C, $F_{1,55} = 0.0003682$, $p = 0.9848$; two-way ANOVA).

Lastly, we examined the effect of heterozygous *Gpr55* deletion on survival of F1.*Scn1a*$^{+/-}$ mice. Dravet syndrome patients and F1.*Scn1a*$^{+/-}$ mice have a significantly reduced lifespan. Sex did not affect survival so males and females were combined. Survival of male and female F1.*Scn1a*$^{+/-}$ mice to P30 was 39% (Fig 3D). This percent survival was not different from male and female F1.*Scn1a*$^{+/-}$;*Gpr55*$^{+/-}$ mice, which was 41% ($p = 0.8817$).

## Pharmacological inhibition of GPR55 does not affect a hyperthermia-induced seizure in *Scn1a*$^{+/-}$ mice

Currently selective GPR55 antagonists, such as CID16020064, are not brain penetrant so in order to evaluate the anticonvulsant potential of pharmacological GPR55 inhibition, a compound with improved pharmacokinetic parameters would need to be used [3]. In the Image-based HTS for Selective Antagonists of GPR55 bioassay conducted at the Sanford-Burnham Center of Chemical Genomics, CID2921524 was identified as a potent GPR55 antagonist with an $IC_{50}$ value of 965 nM (PubChem AID 2013). We first characterized the pharmacokinetic parameters of CID2921524 in mouse plasma and brain. CID2921524 was rapidly absorbed into both plasma and brain following i.p. administration and had relatively long half-lives in

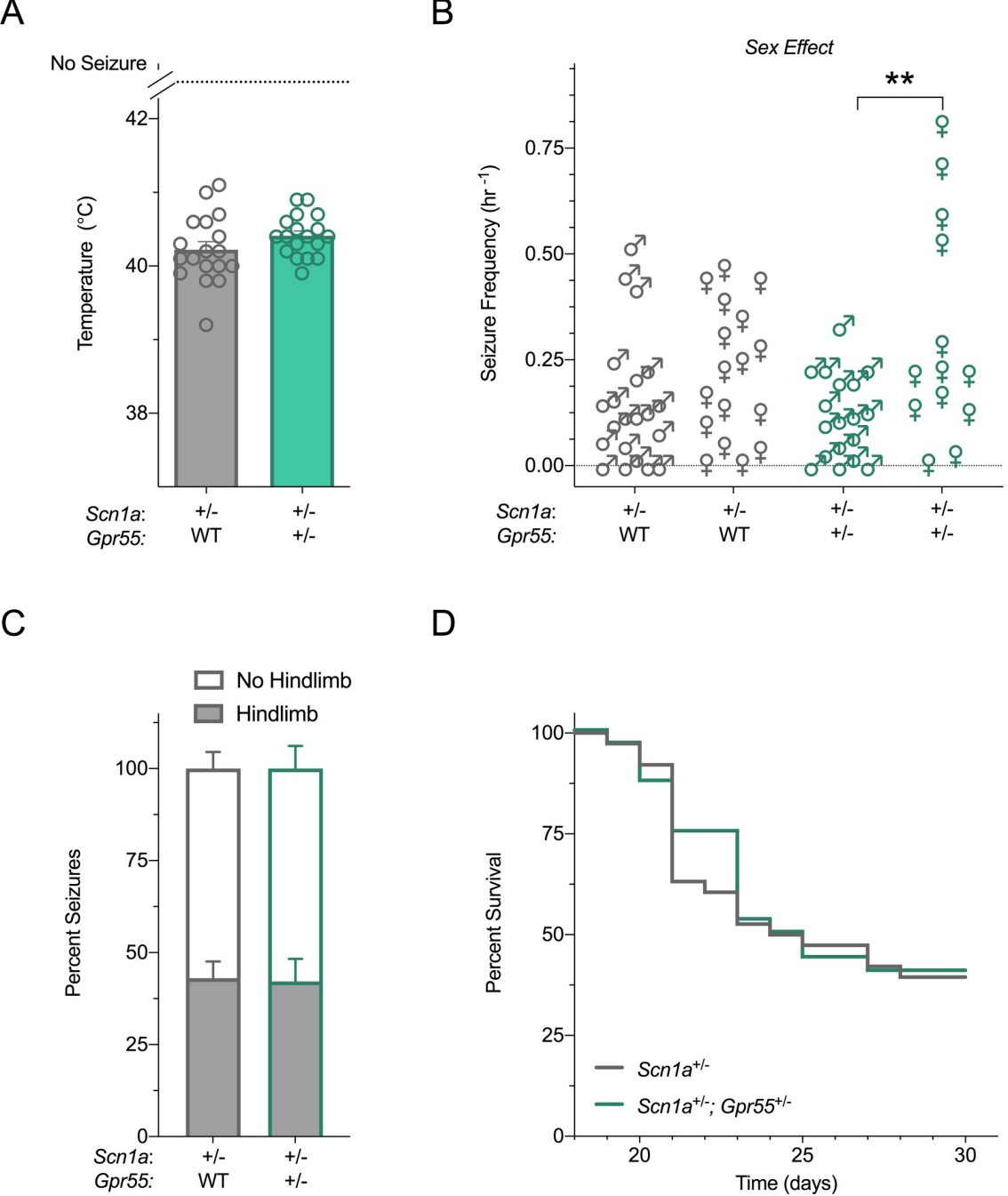

**Fig 3. Genetic deletion of *Gpr55* has little effect on F1.*Scn1a*⁺/⁻ phenotype.** (**A**) Threshold temperature of individual mice for GTCS induced by hyperthermia in male and female F1.*Scn1a*⁺/⁻ (gray symbols) and F1.*Scn1a*⁺/⁻;*Gpr55*⁺/⁻ (green symbols) mice. Heterozygous *Gpr55* deletion had no effect on a thermally induced seizure of F1.*Scn1a*⁺/⁻ mice. The average temperatures of seizure induction are depicted by the bars and error bars represent SEM, with n = 18 (9 males and 9 females) per group (Mantel-Cox logrank). (**B**) Spontaneous GTCS frequency of individual F1.*Scn1a*⁺/⁻ (gray symbols) and F1.*Scn1a*⁺/⁻;*Gpr55*⁺/⁻ (green symbols) mice. Unprovoked, spontaneous GTCS were quantified over a 60 h recording period, with n = 13–21 per group (n = 21, 17, 18, and 13, left to right). A sex-dependent effect was observed with female mice exhibiting a significantly higher spontaneous seizure frequency than male mice ($p < 0.005$, two-way ANOVA). Female F1.*Scn1a*⁺/⁻;*Gpr55*⁺/⁻ mice had a greater spontaneous seizure frequency than male F1.*Scn1a*⁺/⁻;*Gpr55*⁺/⁻ mice (**$p < 0.01$, two-way ANOVA followed by Sidak's test). (**C**) Proportion of spontaneous GTCS with (gray bars) or without (white bars) full tonic hindlimb extension is presented. Heterozygous *Gpr55* deletion had no effect on spontaneous seizure severity of F1.*Scn1a*⁺/⁻ mice (two-way ANOVA). (**D**) Survival curves comparing F1.*Scn1a*⁺/⁻ and F1.*Scn1a*⁺/⁻;*Gpr55*⁺/⁻ male and female mice. Heterozygous *Gpr55* deletion had no effect on survival of F1.*Scn1a*⁺/⁻ mice (Mantel-Cox logrank).

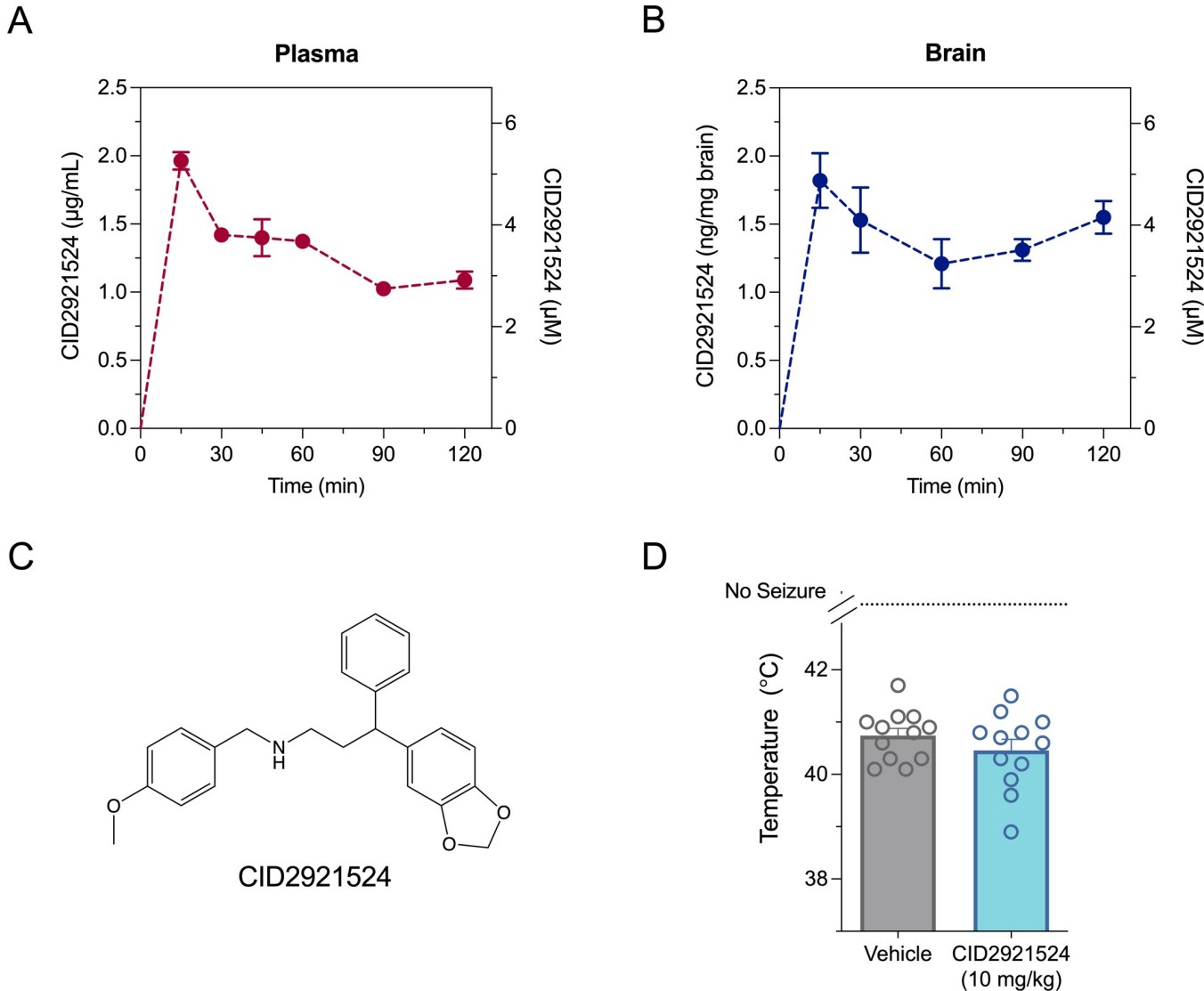

**Fig 4. The GPR55 antagonist CID2921524 is brain penetrant but does not affect hyperthermia-induced seizures in *Scn1a*$^{+/-}$ mice.** Concentration-time curves for CID2921524 in (**A**) plasma and (**B**) brain following a 10 mg/kg i.p. injection. Concentrations are depicted as both mass concentrations (left-axis) and molar concentrations (right y-axis). Data are expressed as means ± SEM, with n = 4 per time point. (**C**) Chemical structure of CID2921524. (**D**) Threshold temperature of individual mice for GTCS induced by hyperthermia in male and female F1.*Scn1a*$^{+/-}$ following acute treatment with vehicle (gray symbols) or 10 mg/kg CID2921524 (blue symbols). CID2921524 had no effect on a thermally induced seizure of F1.*Scn1a*$^{+/-}$ mice. The average temperatures of seizure induction are depicted by the bars and error bars represent SEM, with n = 12 (3 males and 9 females) per group (Mantel-Cox logrank).

both compartments (Fig 4). Total exposure of CID2921524 in brain tissue was greater than that of plasma as determined by AUC values (brain-plasma ratio 1.4). CID2921524 was then evaluated for efficacy against a thermally induced seizure in F1.*Scn1a*$^{+/-}$ mice. CID2921524 had no effect on a hyperthermia-induced seizure (Fig 4D).

## Discussion

Here we sought to examine whether the orphan receptor Gpr55 could be a genetic modifier and potential novel drug target in the *Scn1a*$^{+/-}$ mouse model of Dravet syndrome. We found that seizure-susceptible [129 x B6]F1 mice had significantly elevated *Gpr55* mRNA expression

in the cortex and hippocampus compared to seizure-resistant 129 mice, suggesting Gpr55 might be a genetic modifier. Further, in the seizure-susceptible [129 x B6]F1 strain, mice with heterozygous deletion of *Scn1a* had selectively increased *Gpr55* expression in the cortex when compared to wildtype mice. Thus, we examined the functional implications of these observations by investigating the impact of heterozygous genetic deletion of *Gpr55* on the epilepsy phenotype of F1.*Scn1a*$^{+/-}$ mice. Heterozygous deletion of *Gpr55* did not influence a hyperthermia-induced seizure, spontaneous seizures or survival of F1.*Scn1a*$^{+/-}$ mice.

Our findings are inconsistent with prior data showing that selective pharmacological antagonism or genetic deletion of Gpr55 reduced neuronal excitability in hippocampal slices [3,11]. Sylantyev *et al* (2013) showed that GPR55 agonists increased calcium store discharges and miniature excitatory postsynaptic currents (mEPSCs) in pyramidal cells of hippocampal slices from wildtype mice. The mEPSCs were not, however, evoked by agonists in GPR55 knockout mice [11]. Kaplan *et al* (2017) showed that application of the selective GPR55 antagonist CID16020064 reduced action potential frequency and increased spontaneous inhibitory postsynaptic currents (IPSCs) in current-clamp recordings of dentate granule cells from *Scn1a*$^{+/-}$ mice [3]. While both of these *ex vivo* studies suggest anti-seizure effects of GPR55 inhibition, our study is the first to examine whether Gpr55 affects behavioral seizures *in vivo*.

Heterozygous deletion of *Gpr55* did not impact seizure susceptibility in *Scn1a*$^{+/-}$ mice but this does not preclude the possibility of homozygous deletion being able to yield anticonvulsant effects. Using the breeding strategy we employed here, we were unable to delete both *Gpr55* alleles in F1.*Scn1a*$^{+/-}$ mice. Homozygous *Gpr55* deletion could be achieved by backcrossing F1.*Scn1a*$^{+/-}$;*Gpr55*$^{+/-}$ mice with *Gpr55*$^{+/-}$ mice. However, the ability to resolve the effects of homozygous *Gpr55* deletion would be complicated by the mixed genetic background of the N1 generation. Other methods to delete Gpr55 expression in F1.*Scn1a*$^{+/-}$ mice could be explored in the future using intracerebroventricular or hippocampus-specific viral knockdown of *Gpr55*; however, it would be necessary to ensure complete knockout of *Gpr55* expression. Current selective GPR55 antagonists, such as CID16020064, are not brain-penetrant so could not be evaluated for anticonvulsant efficacy [3]. The GPR55 antagonist CID2921524 was found to have substantial brain uptake but did not have any effect on hyperthermia-induced seizures. While the total concentration of CID2921524 achieved in the brain (~5 μM) is well above its IC$_{50}$ value (965 nM) at GPR55, the free-fraction is unknown. If CID2921524 has high protein binding, then the lack of effect on a hyperthermia-induced seizure could be the result of insufficient concentrations to elicit substantial Gpr55 inhibition. Moreover, CID2921524 is an antagonist of GPR55 but its selectivity for this receptor has not been confirmed. The future development of a brain-penetrant, selective GPR55 antagonist will enable a more rigorous assessment of this class of compound in mouse models of intractable epilepsy. Testing of a small molecule antagonist would provide a more feasible strategy for clinical translation, as opposed to complete genetic deletion of the receptor.

A noteworthy observation of the present study was that heterozygous deletion of *Scn1a* increased mRNA expression of Gpr55 selectively in the cortex of seizure-susceptible [129 x B6] F1 mice. This provides the first evidence of a direct link between Na$_v$1.1 deletion and Gpr55. Future studies could examine the molecular and cellular basis for this interaction. Our results imply that the link may not have any functional significance, at least for hyperthermia-induced seizures, spontaneous seizures and survival of F1.*Scn1a*$^{+/-}$ mice. *Scn1a*$^{+/-}$ mice also exhibit cognitive and behavioral deficits mimicking developmental delays observed in Dravet syndrome patients [3,29–31]. A future study could examine whether Gpr55 contributes to these behavioural phenotypes. Interestingly, the GPR55 agonist O-1602 was recently reported to reduce cognitive deficits, pro-inflammatory cytokines and synaptic dysfunction in a mouse model of Alzheimer's disease [32]. Other studies suggest GPR55 agonists were neuroprotective against

neuroinflammation; whereas, a prolonged inflammatory response was observed in $Gpr55^{-/-}$ mice [16]. Neuroinflammation has not been specifically examined in the $Scn1a^{+/-}$ mice studied here, but recent research suggests that neuroinflammation occurs in the Syn-Cre/Scn1a$^{WT/A1783V}$ mouse model of Dravet syndrome, as evidenced by increased microgliosis, astrogliosis and expression of the proinflammatory cytokine, TNFα [33]. Thus, it is possible that the increased cortical $Gpr55$ expression observed in F1.$Scn1a^{+/-}$ mice serves a neuroprotective rather than a pathogenic role, which could be explored in future studies.

Our results are inconsistent with the view that the anticonvulsant mechanism of action for CBD in Dravet syndrome is solely via blockade of GPR55. There are many additional modes of action that could be involved in the anti-seizure effects of CBD. For example, we have shown that CBD behaves as a positive allosteric modulator (PAM) of GABA$_A$ receptors [34,35], which could explain inhibition of neuronal excitability and reduced seizures. This mechanism is also shared by the phytocannabinoid cannabigerolic acid, which we recently showed to display anticonvulsant properties in $Scn1a^{+/-}$ mice [36]. The transient receptor potential vanilloid type 1 (TRPV1) was shown to mediate the anti-seizure effects of CBD in the maximal electroshock model [37], and heterozygous deletion of $Trpv1$ was proconvulsant against hyperthermia-induced seizures in $Scn1a^{+/-}$ mice [38]. Recent evidence also suggests that CBD might work through the transcription factor, peroxisome proliferator-activated receptor gamma (PPARγ), as the anti-seizure effects of CBD was associated with increased hippocampal PPARγ expression in a rat model of temporal lobe epilepsy [39]. It may also be that the anti-seizure effect of CBD is not mediated by interaction with one drug target, but that it requires a simultaneous action on a variety of targets.

Our study does have limitations, including that we could not ascertain the effects of homozygous deletion of $Gpr55$ on seizure phenotypes of the $Scn1a^{+/-}$ mouse model of Dravet syndrome. Moreover, our results suggesting $Gpr55$ might be a genetic modifier may be specific to our mouse model and future studies are needed to explore whether this translates to Dravet syndrome patients. Future research is needed using alternative approaches and additional probe drugs to further explore GPR55 as a new drug target for treating Dravet syndrome.

## Supporting information

**S1 Fig. Commercially-available GPR55 antibodies.** Western blot analysis of Gpr55 receptor levels in whole brain membrane preparations from wildtype (WT) and $Gpr55^{-/-}$ (KO) mice using (**A**) ThermoFisher, (**B**) Abcam and (**C**) Cayman Chemical primary GPR55 antibodies (right panels) with β-actin or β-tubulin serving as loading controls (left panels). None of these antibodies appear to be selective for mouse Gpr55. Precision Plus Protein Kaleidoscope ladder (Bio-Rad Laboratories). (**D**) Western blot analysis using the Cayman chemical GPR55 antibody blocked with a GPR55 blocking peptide.
(PDF)

**S1 Data.**
(XLSX)

**S1 File.**
(DOCX)

## Acknowledgments

The authors gratefully acknowledge Barry and Joy Lambert for their continued support of the Lambert Initiative for Cannabinoid Therapeutics. The authors thank Samantha Duarte for

technical assistance. We also acknowledge the Monash Genome Modification Platform, Monash University for developing the *Gpr55* knockout mouse line.

## Author Contributions

**Conceptualization:** Samuel D. Banister, Jennifer A. Kearney, Jonathon C. Arnold.

**Data curation:** Lyndsey L. Anderson, Dilara A. Bahceci, Jonathon C. Arnold.

**Formal analysis:** Lyndsey L. Anderson, Dilara A. Bahceci, Nicole A. Hawkins, Declan Everett-Morgan, Jonathon C. Arnold.

**Funding acquisition:** Jennifer A. Kearney, Jonathon C. Arnold.

**Investigation:** Lyndsey L. Anderson, Dilara A. Bahceci, Nicole A. Hawkins, Declan Everett-Morgan, Jennifer A. Kearney, Jonathon C. Arnold.

**Methodology:** Lyndsey L. Anderson, Dilara A. Bahceci, Nicole A. Hawkins, Samuel D. Banister, Jonathon C. Arnold.

**Project administration:** Lyndsey L. Anderson, Jennifer A. Kearney, Jonathon C. Arnold.

**Resources:** Samuel D. Banister, Jennifer A. Kearney, Jonathon C. Arnold.

**Supervision:** Lyndsey L. Anderson, Jennifer A. Kearney, Jonathon C. Arnold.

**Validation:** Lyndsey L. Anderson, Dilara A. Bahceci, Nicole A. Hawkins, Jonathon C. Arnold.

**Visualization:** Jonathon C. Arnold.

**Writing – original draft:** Lyndsey L. Anderson, Jonathon C. Arnold.

**Writing – review & editing:** Lyndsey L. Anderson, Dilara A. Bahceci, Nicole A. Hawkins, Declan Everett-Morgan, Samuel D. Banister, Jennifer A. Kearney, Jonathon C. Arnold.

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
