## [Decision Letter · Decision Letter 0]

25 Oct 2022

PONE-D-22-26036Heterozygous deletion of Gpr55 does not affect a hyperthermia-induced seizure, spontaneous seizures or survival in the Scn1a+/- mouse model of Dravet syndromePLOS ONE

Dear Dr. Arnold,

Thank you for submitting your manuscript to PLOS ONE. After careful consideration, we feel that it has merit but does not fully meet PLOS ONE’s publication criteria as it currently stands. Therefore, we invite you to submit a revised version of the manuscript that addresses the points raised during the review process.

We look forward to receiving your revised manuscript.

Kind regards,

Giuseppe Biagini, MD

Academic Editor

PLOS ONE

Journal Requirements:

"The authors declare that the research was conducted in the absence of any commercial or financial relationships that could be construed as a potential conflict of interest."

"The authors gratefully acknowledge Barry and Joy Lambert for their continued support of the Lambert

Initiative for Cannabinoid Therapeutics. The authors thank Samantha Duarte for technical assistance. We

also acknowledge the Monash Genome Modification Platform, Monash University as a node of

Phenomics Australia, which is supported by the Australian Government Department of Education through

the National Collaborative Research Infrastructure Strategy, the Super Science Initiative and

Collaborative Research Infrastructure Scheme."

"This research was supported by the Lambert Initiative for Cannabinoid Therapeutics, a philanthropically-funded centre for medicinal cannabis research at the University of Sydney, the Australian National Health and Medical Research Council (GNT1161571) and the U.S. National Institutes of Health (R01 NS084959)."

Reviewers' comments:

Reviewer's Responses to Questions

**Comments to the Author**

1. Is the manuscript technically sound, and do the data support the conclusions?

Reviewer #1: Partly

Reviewer #2: Partly

2. Has the statistical analysis been performed appropriately and rigorously? 

Reviewer #1: N/A

Reviewer #2: Yes

3. Have the authors made all data underlying the findings in their manuscript fully available?

Reviewer #1: Yes

Reviewer #2: Yes

4. Is the manuscript presented in an intelligible fashion and written in standard English?

Reviewer #1: Yes

Reviewer #2: Yes

5. Review Comments to the Author

Reviewer #1: The article is of interest. However, in my opinion, some improvements could be made. For instance, I think you could deeply discuss your results. Indeed, you should also take into consideration that perhaps other molecular pathways could be involved in the effects of CBD. Notably, the antiseizure effects of CBD (120 mg/kg i.p.) were associated with upregulation of PPARγ in the hippocampus of a rat model of temporal lobe epilepsy (Costa et al., 2022). Like CBD, cannabigerol (CBG) and tetrahydrocannabinolic acid (Δ9-THCA) have been reported to mediate their anti-inflammatory effects through PPARγ (Stone et al., 2020; Valdeolivas et al., 2015). In addition to this, the upregulation of PPARγ in the hippocampus was not only related to the antiseizure effects of CBD but also to those of EP-80317, a ghrelin receptor antagonist (Lucchi et al., 2017).

Other comments:

1. “…that co-administration of CID16020064 with CBD occluded the effects of CBD (3).” This sentence is unclear. Please rephrase.

2. The company purchasing “CID16020064” and specific information about this compound should be provided.

3. The total number of animals (and the total number of animals per group) should be clarified.

4. The total number of male and female mice per group should be clarified.

5. The statistical test used in the manuscript should be summarized in a specific paragraph.

References:

Lucchi, C.; Costa, A.M.; Giordano, C.; Curia, G.; Piat, M.; Leo, G.; Vinet, J.; Brunel, L.; Fehrentz, J.-A.; Martinez, J.; et al. Involvement of PPARγ in the Anticonvulsant Activity of EP-80317, a Ghrelin Receptor Antagonist. Front. Pharmacol. 2017, 8, 676.

Costa, A.-M.; Russo, F.; Senn, L.; Ibatici, D.; Cannazza, G.; Biagini, G. Antiseizure Effects of Cannabidiol Leading to Increased Peroxisome Proliferator-Activated Receptor Gamma Levels in the Hippocampal CA3 Subfield of Epileptic Rats. Pharmaceuticals 2022, 15, 495. https://doi.org/10.3390/ph15050495

Stone, N. L., Murphy, A. J., England, T. J., & O’Sullivan, S. E. (2020). A systematic review of minor phytocannabinoids with promising neuroprotective potential. British Journal of Pharmacology, bph.15185. https://doi.org/10.1111/bph.15185

Valdeolivas, S., Navarrete, C., Cantarero, I., Bellido, M. L., Muñoz, E., & Sagredo, O. (2015). Neuroprotective properties of cannabigerol in Huntington’s disease: Studies in R6/2 mice and 3-nitropropionate-lesioned mice. Neurotherapeutics: The Journal of the American Society for Experimental NeuroTherapeutics, 12(1), 185–199. https://doi.org/10.1007/s13311-014-0304-z

Reviewer #2: Anderson et al report the effects of a heterozygous deletion of Gpr55 on hyperthermia-induced and spontaneous seizures in the Scn1a+/- mouse model of Dravet syndrome. The experiments are very well performed, and the results are important and interesting. I Have a few minor comments to the authors:

Introduction/Discussion.

The last two sentences of the first paragraph are somewhat redundant.

Although the Scn1a+/- mice show a severe Dravet syndrome phenotype, the effect is strain-specific. Thus, it is likely that it is species-specific too, meaning that this model may not reflect real mechanisms in humans (different genetic modifiers in different species and strains).

Deletion of Gpr55 is not the same as its inhibition, which can be partial, that is maintaining the interaction at a “normal”, non-epileptic level.

These caveats are common for most of studies, but I suggest discussing them. Taking them into account, I also suggest to moderate the main conclusion of the study concerning the antiepileptic mechanisms of cannabinoids (“Our study suggests developing GPR55 antagonists might not be a viable strategy for advancing new treatments”). “to unequivocally rule out GPR55 as a new drug target for treating Dravet syndrome” may not be a goal for future studies. Might be on the contrary?

Methods/Results.

CID2921524: Please, indicate the volume of injected solution per body weight.

This compound reaches high brain levels 15 min after administration. It is not clear at which time point the seizure thresholds were measured. In the description we can read: “mice received a single i.p. injection of vehicle or CID2921524 following the 5 min acclimation period”. Just two or four minutes could be insufficient to penetrate; please explain.

How clonic convulsions were defined?

During GTCS monitoring, were animals housed in groups or individually?

Could the seizure severity be compared by using nonparametric statistics?

How the brains were dissected: frozen, on ice, etc?

“tissue from 3-4 mice (at least one from each sex) were combined into pooled samples (n = 7-11 pools per group) to isolate total RNA” – this description is not entirely clear.

Where the rabbit anti-CB1 polyclonal antibody was obtained?

“CID2921524 is an antagonist of GPR55 antagonist” – please correct.

6. PLOS authors have the option to publish the peer review history of their article (what does this mean?). If published, this will include your full peer review and any attached files.

Reviewer #1: No

Reviewer #2: No

---

## [Author Response · Author response to Decision Letter 0]

29 Nov 2022

REVIEWERS' COMMENTS:

Reviewer #1: The article is of interest. However, in my opinion, some improvements could be made. 

For instance, I think you could deeply discuss your results. Indeed, you should also take into consideration that perhaps other molecular pathways could be involved in the effects of CBD. Notably, the antiseizure effects of CBD (120 mg/kg i.p.) were associated with upregulation of PPARγ in the hippocampus of a rat model of temporal lobe epilepsy (Costa et al., 2022). Like CBD, cannabigerol (CBG) and tetrahydrocannabinolic acid (Δ9-THCA) have been reported to mediate their anti-inflammatory effects through PPARγ (Stone et al., 2020; Valdeolivas et al., 2015). In addition to this, the upregulation of PPARγ in the hippocampus was not only related to the antiseizure effects of CBD but also to those of EP-80317, a ghrelin receptor antagonist (Lucchi et al., 2017).

RESPONSE: We have added a paragraph to the discussion as suggested which considers other molecular pathways of CBD.

Other comments:

1. “…that co-administration of CID16020064 with CBD occluded the effects of CBD (3).” This sentence is unclear. Please rephrase.

RESPONSE: The sentence has been clarified.

2. The company purchasing “CID16020064” and specific information about this compound should be provided.

RESPONSE: We did not use CID16020064 in our study and hence did not purchase the compound. The compound is well-known to be a GPR55 receptor antagonist; we have now made reference to the original paper that showed it was a selective GPR55 antagonist:

Kargl J, Brown AJ, Andersen L, Dorn G, Schicho R, Waldhoer M, & Heinemann A (2013). A selective antagonist reveals a potential role of G protein-coupled receptor 55 in platelet and endothelial cell function. J Pharmacol Exp Ther 346: 54-66.

3. The total number of animals (and the total number of animals per group) should be clarified.

RESPONSE: We have added the number of male and female per group and the totals can now be calculated (see below). Our graphs already contain individual data points.

4. The total number of male and female mice per group should be clarified.

RESPONSE: Numbers of male and female mice per experimental group have been added to Figure Legends.

5. The statistical test used in the manuscript should be summarized in a specific paragraph.

RESPONSE: The manuscript has been corrected accordingly.

Reviewer #2: Anderson et al report the effects of a heterozygous deletion of Gpr55 on hyperthermia-induced and spontaneous seizures in the Scn1a+/- mouse model of Dravet syndrome. The experiments are very well performed, and the results are important and interesting. I Have a few minor comments to the authors:

Introduction/Discussion.

The last two sentences of the first paragraph are somewhat redundant.

RESPONSE: This has been corrected.

Although the Scn1a+/- mice show a severe Dravet syndrome phenotype, the effect is strain-specific. Thus, it is likely that it is species-specific too, meaning that this model may not reflect real mechanisms in humans (different genetic modifiers in different species and strains). Deletion of Gpr55 is not the same as its inhibition, which can be partial, that is maintaining the interaction at a “normal”, non-epileptic level. These caveats are common for most of studies, but I suggest discussing them. 

RESPONSE: We have added some discussion of these points as requested.

Taking them into account, I also suggest to moderate the main conclusion of the study concerning the antiepileptic mechanisms of cannabinoids (“Our study suggests developing GPR55 antagonists might not be a viable strategy for advancing new treatments”). “to unequivocally rule out GPR55 as a new drug target for treating Dravet syndrome” may not be a goal for future studies. Might be on the contrary?

RESPONSE: This conclusion has been moderated as suggested.

Methods/Results.

CID2921524: Please, indicate the volume of injected solution per body weight.

RESPONSE: This has been corrected.

This compound reaches high brain levels 15 min after administration. It is not clear at which time point the seizure thresholds were measured. In the description we can read: “mice received a single i.p. injection of vehicle or CID2921524 following the 5 min acclimation period”. Just two or four minutes could be insufficient to penetrate; please explain.

RESPONSE: Mice received an injection of vehicle or CID2921524 immediately prior to initiation of the hyperthermia protocol (body temperature elevated 0.5 °C every 2 min until the onset of first clonic convulsion with loss of posture). On average, the hyperthermia protocol takes roughly 15 - 25 mins. The experiment was designed so that CID2921524 brain concentrations are near Cmax, when core body temperatures reach 40.5 - 41 °C.

How clonic convulsions were defined?

RESPONSE: The clonic convulsions for hyperthermia-induced seizures were defined as the first clonic spasm of the forelimbs and/or hindlimbs with loss of posture.

During GTCS monitoring, were animals housed in groups or individually?

RESPONSE: Mice were group housed during. Methods have been updated.

Could the seizure severity be compared by using nonparametric statistics?

RESPONSE: Seizure severity was compared using ANOVA as the data were normally distributed. This analysis is appropriate and consistent with our prior published data using this model. 

How the brains were dissected: frozen, on ice, etc?

RESPONSE: Mice were sacrificed at P24 by cervical dislocation and immediately decapitated. Whole brains were extracted and frozen immediately with liquid nitrogen and stored at -80C. For dissections, whole brain samples were thawed on ice before the hippocampi from each hemisphere was extracted and frozen with liquid nitrogen and stored at -80 C. Methods have been updated. 

“tissue from 3-4 mice (at least one from each sex) were combined into pooled samples (n = 7-11 pools per group) to isolate total RNA” – this description is not entirely clear.

RESPONSE: The methods section has been updated to clarify the pooled samples:

Briefly, cortex and hippocampi were dissected from mice at postnatal day 24 (P24) and tissue from 3-4 mice (at least one from each sex) were combined into pooled samples to isolate total RNA. Cortical and hippocampal samples were dissected and pooled from different cohorts of F1.Scn1a+/- mice. First strand cDNA was synthesized (Superscript IV; Life Technologies; Carlsbad, CA, USA) from 2 µg total RNA and ddPCR was performed using ddPCR Supermix for Probes (No dUTP; Bio-Rad; Hercules, CA, USA) and TaqMan Assays as previously described [23] on n = 7-11 pooled samples per group. Taqman gene expression assays (Life Technologies) were mouse Gpr55 (FAM-MGB-Mm02621622_s1) and Tbp (VIC-MGB-Mm00446971_m1). Relative transcript levels were expressed as a concentration ratio to Tbp.

Where the rabbit anti-CB1 polyclonal antibody was obtained?

RESPONSE: The anti-CB1 antibody was incorrectly entered. This has been removed from the manuscript.

“CID2921524 is an antagonist of GPR55 antagonist” – please correct.

RESPONSE: This has been corrected.

---

## [Decision Letter · Decision Letter 1]

10 Jan 2023

Heterozygous deletion of Gpr55 does not affect a hyperthermia-induced seizure, spontaneous seizures or survival in the Scn1a+/- mouse model of Dravet syndrome

PONE-D-22-26036R1

Dear Dr. Arnold,

We’re pleased to inform you that your manuscript has been judged scientifically suitable for publication and will be formally accepted for publication once it meets all outstanding technical requirements.

Kind regards,

Giuseppe Biagini, MD

Academic Editor

PLOS ONE

Additional Editor Comments (optional):

Reviewers' comments:

Reviewer's Responses to Questions

**Comments to the Author**

1. If the authors have adequately addressed your comments raised in a previous round of review and you feel that this manuscript is now acceptable for publication, you may indicate that here to bypass the “Comments to the Author” section, enter your conflict of interest statement in the “Confidential to Editor” section, and submit your "Accept" recommendation.

Reviewer #1: All comments have been addressed

Reviewer #2: (No Response)

2. Is the manuscript technically sound, and do the data support the conclusions?

Reviewer #1: Yes

Reviewer #2: (No Response)

3. Has the statistical analysis been performed appropriately and rigorously? 

Reviewer #1: Yes

Reviewer #2: (No Response)

4. Have the authors made all data underlying the findings in their manuscript fully available?

Reviewer #1: Yes

Reviewer #2: (No Response)

5. Is the manuscript presented in an intelligible fashion and written in standard English?

Reviewer #1: Yes

Reviewer #2: (No Response)

6. Review Comments to the Author

Reviewer #1: All comments have been addressed.

Reviewer #2: (No Response)

7. PLOS authors have the option to publish the peer review history of their article (what does this mean?). If published, this will include your full peer review and any attached files.

Reviewer #1: No

Reviewer #2: No

---

## [Editor Report · Acceptance letter]

19 Jan 2023

PONE-D-22-26036R1 

Heterozygous deletion of Gpr55 does not affect a hyperthermia-induced seizure, spontaneous seizures or survival in the Scn1a+/- mouse model of Dravet syndrome 

Dear Dr. Arnold:

I'm pleased to inform you that your manuscript has been deemed suitable for publication in PLOS ONE. Congratulations! Your manuscript is now with our production department. 

Kind regards, 

on behalf of

Dr. Giuseppe Biagini 

Academic Editor

PLOS ONE